# Established Immortalized Cavernous Endothelial Cells Improve Erectile Dysfunction in Rats with Cavernous Nerve Injury

**DOI:** 10.3390/ph16010123

**Published:** 2023-01-13

**Authors:** Sang Hong Bak, Jae Heon Kim, Seung U. Kim, Dong-Seok Lee, Yun Seob Song, Hong J. Lee

**Affiliations:** 1Research Institute, Humetacell Inc., Bucheon 14786, Gyeonggi, Republic of Korea; 2Department of Urology, Soonchunhyang University School of Medicine, Seoul 04401, Republic of Korea; 3Division of Neurology, Department of Medicine, UBC Hospital, University of British Columbia, Vancouver, BC V6T 2B5, Canada; 4BK21 Plus KNU Creative BioResearch Group, School of Life Sciences, Kyungpook National University, Daegu 41566, Republic of Korea; 5School of Life Sciences & Biotechnology, College of Natural Sciences, Kyungpook National University, Daegu 41566, Republic of Korea; 6Medical Research Institute, Chungbuk National University, Cheongju 28644, Chungbuk, Republic of Korea

**Keywords:** erectile dysfunction, endothelial cells, *v-myc*

## Abstract

The main cause of erectile dysfunction (ED) is the damage in penile cavernous endothelial cells (EC). Murine primary ECs have a limited growth potential, and the easy availability of murine ECs will facilitate the study of cavernous endothelial dysfunction in rats. This study was performed to establish immortalized rat penile cavernous ECs (rEC) and investigate how they could repair erectile dysfunction in rats with cavernous nerve injury (CNI). rEC was isolated enzymatically by collagenase digestion and were cultured. An amphotropic replication-incompetent retroviral vector encoding *v-myc* oncogene was used to transfect rEC for immortalization (vREC). Morphological and immunohistochemical properties of vREC were examined. Eight-week-old male Sprague-Dawley rats were divided into three groups of five rats each, including group 1 = sham operation, group 2 = bilateral CN injury, group 3 = vREC (1 × 10^6^ cells) treatment after CNI. Erectile response was assessed at 2, 4 weeks after transplantation of vREC., Penile tissue were harvested at 4 weeks after transplantation and immune–histochemical examination was performed. vREC showed the expression of CD31, vWF, cell type-specific markers for EC by RT-PCR and flowcytometry. At 2, 4 weeks after transplantation, rats with CNI had significantly lower erectile function than control group (*p* < 0.05). The group transplanted with vREC showed higher erectile function than the group without vRECs (*p* < 0.05). vREC was established and repaired erectile dysfunction in rats with CNI. This cell line may be useful for studying mechanisms and drug screening of erectile dysfunction of rats.

## 1. Introduction

Radical prostatectomy is the choice of treatment for localized prostate cancer. Post-prostatectomy erectile dysfunction typically results from the cavernous nerve injury (CNI) that runs along the posterolateral aspects of the prostate and provides most of the autonomic input to erectile tissue [1]. However, erectile dysfunction (ED) is still a major complication of the procedure even with the nerve sparing approach [2]. Although phosphodiesterase type 5 inhibitor therapy is a used for the treatment of ED following bilateral nerve-sparing radical prostatectomy, these drugs remain largely inefficient [3]. Recently, cell-based therapy was focused on the repair of ED following CNI [4].

The damage of smooth muscle cells, endothelial cells, and increased collagen deposition of penile cavernous nerve injury happened after CN injury [5]. The main cause of erectile dysfunction (ED) is the damage in cavernous endothelial cells (EC). The transplantation of EC into the cavernosum could offer a chance of recovery for damaged ECs.

Murine ECs may be useful for studying mechanisms and drug screening of erectile dysfunction of rats.

Murine primary ECs have a more limited growth potential, and the easy availability of murine ECs will facilitate the study of cavernous endothelial dysfunction in rats. Previously, we have isolated clonal human neural stem cell lines that had been immortalized by a retroviral vector encoding *v-myc* oncogene [6,7,8,9]. To obtain large amounts of murine EC, their immortalization has been strongly recommended. To obtain a suitable suspension of uncontaminated murine endothelial cells in adequate quantities, isolation of a clonal rat primary EC and their immortalization using a retroviral vector encoding the *v-myc* oncogene was performed. In this study, we established immortalized rat penile cavernous ECs (rEC) and investigated their usefulness after their transplantation into the cavernosum of rats with cavernous nerve injury.

## 2. Results

### 2.1. Immortalization of Penile Cavernous Endothelial Cells

The established *v-myc* immortalized rECs showed the expression of *v-myc* transcript by reverse transcription-polymerase chain reaction (RT-PCR) (Figure 1B). Culture of penile cavernous endothelial cells reaching confluence after 7–10 days showed a “cobblestone” morphology. They were free of contaminating spindle cells (Figure 1C). Sterile sorting for CD31-positive cells ensured 95% rECs and reduced non-endothelial contaminating cells. Immortalized penile cavernous endothelial cells (vRECs) of rats are shown (Figure 1D). They continued to proliferate to more than 70 population doublings and did not undergo a crisis. vRECs are isolated by flowcytometry and most of the vRECs express the endothelial marker (CD31) (Figure 2).

### 2.2. RT-PCT of vRECs

The endothelial markers (CD31, CD105, vWF, Flt-1, KDR, GAPDH) are well expressed in vREC. (Figure 3).

### 2.3. In Vitro Angiogenesis Assay of Immortalization of Penile Cavernous Endothelial Cells

vRECs demonstrated the tube formation which created a complex three-dimensional capillary network, suggesting the morphological properties of primary EC (Figure 4A,B), and capillary network looks firmer between vRECs than between rECs (Figure 4C,D).

### 2.4. Change of Collagen Deposition after Transplantation

The mean percent collagen area of corpus cavernosa in group 1 was 7.4 ± 1.1. It increased in group 2 (14.3 ± 1.4) compared with group 1 (*p* < 0.05) and recovered in group 3 (9.6 ± 1.1) compared with group 2 at 2 weeks (*p* < 0.05). It increased in group 2 (13.7 ± 0.2) compared with group 1 and recovered in group 3 (8.8 ± 3.0) compared with group 2 at 4 weeks (*p* < 0.05) (Figure 5).

### 2.5. Erectile Response to Nerve Stimulation in Cavernous Nerve Injury Model of Rats

At 2, 4 weeks, the group with cavernous nerve injury showed significantly lower ICP/MAP than the group without cavernous nerve injury (*p* < 0.05). The group transplanted with vREC injection showed higher ICP/MAP than the group without vREC injection (*p* < 0.05) (Figure 6).

### 2.6. Immunohistochemical Study

At 4 weeks, the group with cavernous nerve injury showed decreased density of expression of cavernous endothelial and smooth muscle makers than the sham operation group. Transplantation of vREC showed increased expression of cavernous endothelial (vWF) and smooth muscle makers (αSMA) than the cavernous nerve injury group (Figure 7).

## 3. Materials and Methods

### 3.1. Preparation of Rat Cavernous Endothelial Cells

All procedures were conducted in accordance with the National Institute of Health Guide for the Care and Use of Laboratory Animals and were approved by the Institutional Animal Care and Use Committee of our hospital. Ten-week-old male Sprague Dawley rats (300–320 g, *n* = 5) were used in this study. A portion of the penis between the decussation of the crura proximally and the glans penis distally was harvested. The cavernous bodies were washed free of blood. Dartos and Buck’s fascia of the penis were removed. The cavernous bodies were then suspended in a flask of saline and transported packed in ice. For collagenase digestion, tissue samples were transported in ice-cold normal saline, minced, and digested in 0.1% collagenase. The digestion was stopped with Dulbecco’s PBS/2% BSA, and the resultant slurry was filtered through a 70 m nylon cell strainer (Becton Dickinson, Franklin Lakes, NJ, USA). The solution was centrifuged at 550× *g* (7 min, 22 °C), the cell pellet was re-suspended in Medium-199 (Gibco BRL, Gaithersburg, MD, USA) with 2% fetal bovine serum (FBS; Gibco, Billings, MT, USA) and EGM-2 SingleQuots™ (Cambrex, East Rutherford, NJ, USA), and was grown on 1% gelatin B (Sigma Chem Co., St Louis, MO, USA)-coated 12-well plates. Cavernosal tissue was placed in sterile PBS/BSA (2 mg/mL), washed gently to remove red blood cells, and minced with sterile scissors. After removing the PBS/BSA, explants were washed once with complete medium and spread with a wide-bore pipette into T75 gelatin-coated flasks. Fresh media (4–5 mL) was then added to cover the explants without allowing them to float. Upon visualization of EC migrating out, the explants were removed, colonies monitored for smooth muscle cell contamination, and EC allowed to proliferate. Cell sorting with PE-conjugated anti-CD31 antibody (1:50; Becton Dickinson) provided >95% CD31-positive cells using fluorescence activated cell sorter (FACS Vantage™, Becton Dickinson). We confirmed the endothelial origin of the cells by uptake of Alexa-488-acetylated-LDL (Molecular Probes, Eugene, OR, USA) and immunofluorescence for CD-31 (BD Pharmingen, San Diego, CA, USA).

### 3.2. Cell Culture

RECs were grown on 2% gelatin type B (Sigma Co., St Louis, MO, USA)-coated tissue culture dishes in EBM™-2 medium (Lonza, Basel, Switzerland) supplemented with EGM™-2 Endothelial Cell Growth Medium-2 BulletKit™ (Lonza). All cells were grown in a humidified incubator at 37 °C and 5% CO_2_ with media changed every 3 days.

### 3.3. Immortalized Rat Cavernous Endothelial Cells

An amphotropic replication-incompetent retroviral vector encoding *v-myc* oncogene (transcribed from mouse leukemia virus LTR plus neomycin-resistant gene transcribed from a CMV promotor) was used to infect rEC inducing propagation of immortalized rEC (Figure 1). This amphotropic vector, LM myc, was generated in our laboratory using the ecotropic retroviral vector encoding *v-myc* (ATCC, Manhasset, VA, USA) to infect PA317 amphotropic packaging cell line. RECs were subjected to retrovirus mediated-transduction of *v-myc* by LMmyc construct and subsequent cloning. Infection of rEC in 6-well plates was performed twice by the established procedures. Briefly, 2 mL of supernatant (4 × 10^5^ CFUs) from the packaging cell line and 8 μg/mL polybrene (Sigma) were added to target cells in 6-well plates and incubated for 4 h at 37 °C; the medium was then replaced with fresh growth medium; infection was repeated 24 h later. Seventy-two hours after the second infection, infected cells were selected with G418 (250 μg/mL, Sigma) for 7–14 days, and large clusters of clonally derived cells were individually isolated and grown in 6-well plates. Individual clones were generated by limited dilution and propagated further. At this phase of isolation, individual clones were designated as rEC.

### 3.4. Confirmation of vREC

Cell lines were isolated by limiting dilution and their clonal identity confirmed by presence on Southern blot of a single retroviral insertion site in all members of the population. RT-PCR confirmed presence of *v-myc* gene.

### 3.5. Reverse Transcriptase PCR (RT-PCR)

Total RNA was extracted using Ribospin™ II (GeneAll^®^, Seoul, Republic of Korea; 314-103) by the manufacturer’s instructions. cDNA was synthesized from 1 µg total RNA by RocketScript™ Reverse Transcriptase Kit (Bioneer, Daejeon, Republic of Korea; E-3141). RT-PCR was conducted using amfiXpand PCR Master Mix (GenDEPOT, Katy, TX, USA; P0331-050) by the manufacturer’s instructions. The whole thermos-cycle processes were performed in Veriti™ 96-Well Thermal Cycler (Applied Biosystems, Waltham, MA, USA; #9902), The used primers are listed on Table 1.

### 3.6. In Vitro Angiogenesis Assay

Cell adherence matrix (Matrigel^®^ Matrix, Corning, NY, USA; Cat #354234) was added to a 24-well plate on ice and incubated for 1 h at 37 °C. The same number of rECs and vRECs were added to the plate (about 1.4 × 10^6^) and incubated for 18 h. Then, the media was eliminated and washed twice with Hank’s balanced salt solution (HBSS, Welgene, Gyeongsan-si, Gyeongsangbuk-do, Republic of Korea; Cat #LB003-03), and 8 μg/mL of Calcein AM fluorescent dye (Corning, #354216) was added to each well, and the cells were incubated for 30 min. Then the cells were washed once by HBSS, and images of the cells were taken under a fluorescence microscope.

### 3.7. Flowcytometry Analysis

Cyflow Cube 8 FACS instrument (SysmexPartec, Görlitz, Germany) was used and data were analyzed using standard FSC Express software (De Novo Software, Los Angeles, CA, USA). The myc immortalized rECs (vRECs) cells were subjected to flow cytometry analysis using anti-CD31 (1:100, Origene Technologies, Rockville, MD, USA) antibody. Two-color flow cytometry (BD FACS Canto II; BD, Franklin Lakes, NJ, USA) was used to analyze the immune-stained cells. By comparing the results with the corresponding negative controls, the percentage of stained cells was calculated.

### 3.8. Transplantation of vREC in Rats with Cavernous Nerve Injury

All procedures were conducted in accordance with the National Institute of Health Guide for the Care and Use of Laboratory Animals (2001) and were approved by the Institutional Animal Care and Use Committee of our hospital. Eight-week-old male Sprague-Dawley rats were divided into three groups of five rats each. Group 1 = sham operation, group 2 = bilateral CN injury, group 3 = vREC (1 × 10^6^ cells) treatment after CN injury. Rats were anesthetized with 1% ketamine (30 mg/kg) and xylazine hydrochloride (4 mg/kg). The penile skin incision was made, and the penile cavernosum was dissected and palpated. vREC (1 × 10^6^ cells) were transplanted into the rat cavernosum using 500 microliters syringe with 26G needle. Before injection, an elastic tourniquet was applied at the base of the penis and kept in place for 2 min, and the incised penile skin was sutured. 10 mg/kg injection of Flomoxef (cephalosporin; Ildong Pharmaceutical Co., LTD., Seoul, Republic of Korea) was administered i.p. daily to prevent infection. The penile cavernosum was harvested 4 weeks after the cell transplantation from groups 1–3.

### 3.9. Measurement of Erectile Function

Erectile response was assessed in groups 1–3 at 4 weeks at 2, 4 weeks after transplantation. Rats were anesthetized intraperitoneally with 1% ketamine (30 mg/kg) and xylazine hydrochloride (4 mg/kg), and the MPG (major pelvic ganglia) and CN (cavernous nerve) were exposed bilaterally via midline laparotomy. The right carotid artery was cannulated to monitor arterial blood pressure.

The penis was exposed, and the corpus spongiosum was mobilized to facilitate insertion of a 25-gauge needle into the corpus cavernosum. The needle was attached via polyethylene-50 tubing to a pressure transducer filled with heparinized saline. For electrically stimulated penile erections, a bipolar platinum electrode was placed around the cavernous nerve. Stimulation parameters were 1 V at a frequency of 12 Hz, with square-wave duration of 1 millisecond for 1 min. The ratio of maximal intracavernous pressure (ICP) to mean arterial pressure (MAP) was determined to control for variations in systemic blood pressure.

### 3.10. Histology

At the end of the erectile function measurement, each animal was anesthetized and perfused through the heart with 100 mL of cold saline and 100 mL of 4% paraformaldehyde in PBS. After 24 h of fixation in 4% paraformaldehyde, the bladder was cryoprotected in 30% sucrose for 24 h, cut into 20 μm sections in a cryostat (Leica, Wetzlar, Germany; CM 1900), and stained with Masson’s trichrome. Each slide was inspected microscopically, and 10 randomly chosen representative areas from light microscope images were captured. Captured video images were then displayed on a color monitor and simultaneously digitized and analyzed using an IBM^®^ computer. To evaluate the results of Masson’s trichrome staining, representative portions of each slide were calculated in a blinded fashion with a square micrometer, and the mean area was expressed as the relative percent. The mean percent collagen area was defined according to the formula, (collagen)/(collagen + muscle). This technique was predicated on the area calculation of the smooth muscle, which was stained red, and connective tissue, which was stained blue. Quantitative image analysis was done using OPTIMAS, version 6.1 (Media-Cybernetics^®^, Rockville, MD, USA).

### 3.11. Immunohistochemistry

At the end of the erectile function study, animals were perfused through the heart with 100 mL of cold saline and 100 mL of 4% paraformaldehyde in PBS. After 24 h of fixation in 4% paraformaldehyde, the penis was cryoprotected with 30% sucrose for 24 h and cut into 20 μm sections in a cryostat (Leica CM 1900). Serial coronal sections were processed for RITC immunofluorescent staining to identify the transplanted rECs. Antibodies specific for alpha-smooth muscle actin (1:200, Abcam, Cambridge, UK; Cat #ab15734), von Willebrand factor (1:100, Abcam, Cat #ab9378) were used for cell type identification of endothelial cells. Penis sections were incubated in primary antibodies overnight at 4 °C as free floating sections and followed by Alexa Fluor 488-conjugated anti-mouse IgG (1:1000, Molecular Probes, Cat #A-11029) and Alexa Fluor 594-conjugated anti-mouse IgG (1:1000, Molecular Probes, Cat #A-11005) for 1 h at room temperature. Negative control sections from each animal were prepared for immunohistochemical staining in an identical manner except the primary antibodies were omitted. Stained sections were then examined under an Olympus laser confocal fluorescence microscope.

### 3.12. Statistical Analysis

Two-way ANOVA and the post-hoc Tukey test were used for analysis of stem cell transplantation. Data are presented as means ± SE, and *p* < 0.05 was considered statistically significant.

## 4. Discussion

We transplanted allogenic *v-myc*-transfected cells into the corpus cavernosum of rodents with CNI. This technique could be used to repair the damaged corporal endothelium and provides the basis for cell-based therapy of erectile dysfunction. Micro-vessel endothelial cells can be isolated in large numbers, proliferate rapidly in cell culture, and allow acceptable transduction efficiencies with retroviral vectors [10,11]. The proximity of the transplanted endothelial cells to trabecular smooth muscle suggests that modulation of the target cell might be feasible, although it does not guarantee functional endothelial–-smooth-muscle interaction [10,11]. The transplanted cells adhere to the sinusoidal endothelium and remain in the penis for up to two weeks. Endothelial cells self-aggregate and incorporate into monolayers. Soon after transplantation, the cells were diffusely distributed and localized near the tunica at 1 and 2 weeks [10,11]. Transplanted cells could become a part of the sinusoidal lining of the penis [11,12].

Previously we have isolated clonal human neural stem cell lines that had been immortalized by a retroviral vector encoding *v-myc* oncogene [6,7,8,9]. Murine ECs may be useful for studying mechanisms and drug screening of erectile dysfunction of rats. Murine primary ECs have a more limited growth potential and the easy availability of murine ECs will facilitate the study of cavernous endothelial dysfunction in rats. In this study, immortalization and cloning of rECs into stable permanent cell lines (vREC) represent our attempt to overcome some of the limitations of primary cultures of EC and provide a potentially significant experimental model for biomedical research. 

To keep them in a proliferative state, we have transduced primary rEC with the retroviral vector encoding immortalizing gene *v-myc*. The stable immortalized cell lines of rEC provide an unlimited number of homogenous EC and facilitate follow up of progeny of the same EC for a prolonged period, over many generations.

The risk of *myc* is its tumor-inducing potential, which offsets its great usefulness in promoting self-renewal. However, if *myc*’s tumor-inducing potential could be considerably deactivated in vivo, it might be the only gene needed to be invoked in vitro to enhance self-renewal of an endothelial progenitor, preserving its multipotency or oligopotency without inducing pluripotency. In previous studies, we have tried to verify the different types of *v-myc*-transfected immortalized cells [13,14]. Some evidence has shown successful differentiation of *v-myc*-transfected cells in rodents [15,16,17], and *v-myc* has been a useful and safe oncogene for immortalized cells without tumorigenesis [18].

The use of endothelial cell cultures provides the advantage of providing a homogeneous cell population but has the disadvantage of low yield after several passages in culture [12,19,20,21,22,23,24]. It is necessary to obtain an appropriate amount of suspended, uncontaminated endothelial cells.

The isolation of the pure population of viable endothelial cells from the cavernous bodies of rat penis is based on collagenase and release of the cells lining the lacunae of the corpora cavernosa (Figure 1). The cells are derived from the complex network of sinusoid endothelium which constitutes the vascular structure of erectile tissue. Identification of the isolated endothelial cells was based on morphological features and on the presence of von Willebrand factor and CD31 which are considered specific markers of endothelial cells.

CD31, as the endothelial cell adherence factor, is known to be involved in angiogenesis [25]. CD105, as known as endoglin, also a remarkable marker for angiogenesis of endothelial cell in various tissues [25,26,27]. vWF (von Willebrand factor) participates in the blood clotting process and is especially used as a marker in penile cavernosal cells [28,29]. VEGFR1/2 (flt1, kdr) are known as a marker of angiogenesis in human penile cavernosal cell [25,28,29,30]. All the markers involved in endothelial angiogenesis were expressed at a relatively high level in *v-myc* transfected established cells compared to normal rat cavernosal endothelial cells.

The endothelial cells were isolated from primary culture by magnetic beads coated by specific antibodies (Figure 2 and Figure 3). To verify phenotype, endothelial cells were stained with antibodies that recognize endothelial cells specific markers. The cells were positively stained CD31 antibodies indicating their endothelial origin. This method allows the preparation of many endothelial cells free from the other components of the vessel wall such as fibroblasts, smooth muscle cells, and from erythrocytes, thus making it possible to conduct functional studies on isolated endothelial cells. Immunohistochemically isolated endothelial cells from corpus cavernosum had endothelial cell-like morphology and formed a cobblestone-like structure when they reached confluence. Sorting for CD31-positive cells ensured > 95% ECs (Figure 2 and Figure 3).

For the in vitro angiogenesis assay, immunohistochemically isolated endothelial cells from corpus cavernosum demonstrated the tube formation which created a complex three-dimensional capillary network, suggesting the morphological properties of ECs (Figure 4).

Ratio of smooth-muscle/collagen decreased in cavernous nerve injury rats and recovered after transplantation of neural-like cells from adipose-derived stem cells with the cavernous nerve injury [5,24,31]. The mean percent collagen area in corpus caversnosum decreased after cavernous nerve injury, but with the transplanted vREC into corpus cavernosa of rats, the mean percent collagen area increased in corpus caversnosa in rats with cavernous nerve injury (Figure 5). Decrease of the mean percent collagen area means decreased collagen or enrichment of smooth muscle after transplantation of vREC. Angiogenesis can mediate protective cellular processes and abrogating the inflammation may be also proposed as mechanisms for the therapeutic effects [31,32].

At 2, 4 weeks after transplantation, the group with cavernous nerve injury had significantly lower ICP/MAP than the group without cavernous nerve injury (Figure 6) (*p* < 0.05). The group transplanted with vREC injection showed higher ICP/MAP than the group without vREC injection (*p* < 0.05). (Figure 6) Improvement of erectile function was found in rats with CNI after the transplantation of vREC in response to cavernous nerve stimulation. At 4 weeks, the group with cavernous nerve injury showed decreased density of expression of cavernous endothelial (vWF) and smooth muscle maker (SMA) than the sham operation group. Transplantation of vREC showed increased expression of cavernous endothelial and smooth muscle makers than the cavernous nerve injury group (Figure 7).

The proposed mechanism for the therapeutic effects is a source for cell replacement of damaged endothelial cells. Angiogenesis and enrichment of smooth muscle of cavernosum were demonstrated [32,33]. Transplantation of endothelial cells induces angiogenesis and promotes protective cellular processes [31,32,33]. Capillary-directed treatment via endothelial progenitor cell therapy stands as a potent approach in abrogating the inflammation-associated secondary cell death of stroke [31,32,33].

Transplanted cells can also secrete growth factors and contribute to regeneration through paracrine mechanisms [34]. Transplanted cells can also secrete growth factors and contribute to reducing fibrosis through paracrine mechanisms rather than cell incorporation [34,35]. They are known to reduce fibrosis in the liver or lung [36,37].

In summary, our results showed the isolation of endothelial cells from the cavernous bodies of the rat penis, and we showed the establishment of the immortalized cavernous endothelial cells by transfection of retroviral *v-myc* oncogene, which has actual endothelial gene expressions in vitro and a recovering function in vivo for impotent from cavernous nerve injury in rats. Such cells are immortalized, and the yield is sufficiently high to permit studies which require large amounts of material.

## 5. Conclusions

Immortalized rat penile cavernous endothelial cells using retroviral vector encoding *v-myc* were established and repaired erectile dysfunction in rats with CNI. This cell line may be useful for studying mechanisms and drug screening of erectile dysfunction of rats.

## Figures and Tables

**Figure 1 pharmaceuticals-16-00123-f001:**
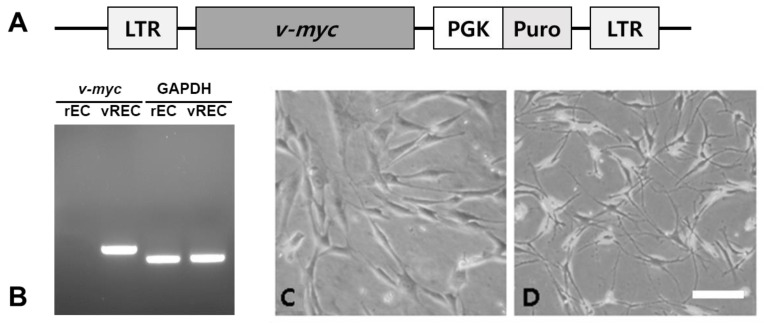
vREC expresses *v-myc* immortalized rat endothelial cell line. (**A**): vREC were infected with a retroviral vector encoding *myc*. (**B**): The expression of *v-myc* transcript from vREC was confirmed but not from rEC by RT-PCR. (**C**,**D**): Phase-contrast microscopic images. (**C**): rEC, (**D**): vREC. rEC = rat cavernous endothelial cells, vREC = immortalized rat cavernous endothelial cells using retroviral vector encoding *v-myc*. Scale bars = 100 μm.

**Figure 2 pharmaceuticals-16-00123-f002:**
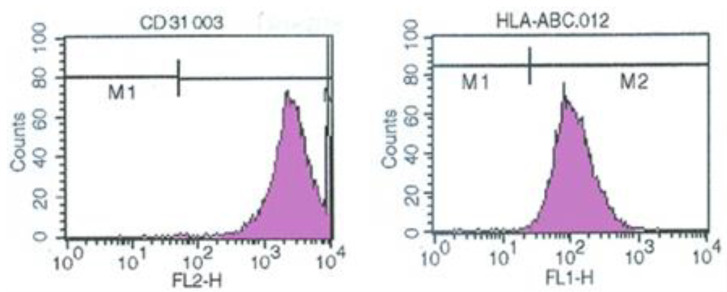
vREC showed the expression of endothelial CD31 cell type-specific markers by flowcytometry. vREC = immortalized rat cavernous endothelial cells using retroviral vector encoding *v-myc*.

**Figure 3 pharmaceuticals-16-00123-f003:**
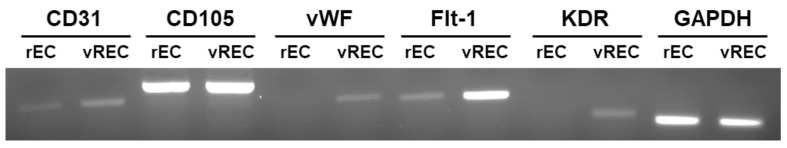
The expression of endothelial cell markers between rEC and vREC by RT-PCR. vWF = von Willebrand factor, Flt-1 = Fms-like tyrosine kinase 1, KDR = kinase insert domain receptor, GAPDH = glyceraldehyde-3-phosphate dehydrogenase, rEC = rat cavernous endothelial cells, vREC = immortalized rat cavernous endothelial cells using retroviral vector encoding *v-myc*.

**Figure 4 pharmaceuticals-16-00123-f004:**
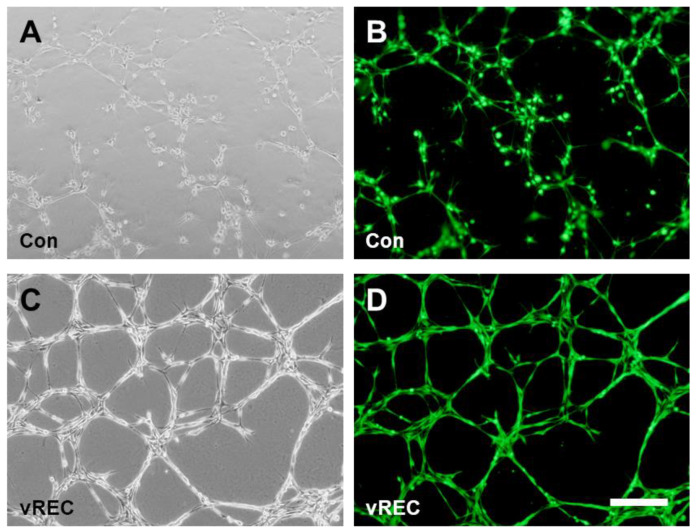
In vitro angiogenesis assay. (**A**,**C**): Phase-contrast microscopic images for angiogenesis assay. (**A**): rECs on Matrigel demonstrated the capillary-like structures suggesting the morphological properties, (**C**): vRECs on Matrigel demonstrated the morphological tubular formation. (**B**,**D**): Immunofluorescent microscopic images for angiogenesis assay. (**B**): rECs, (**D**): vRECs, the cells in the each groups were seeding with the same number for the assay. Con = control cells (rat cavernous endothelial cells), vREC = immortalized rat cavernous endothelial cells using retroviral vector encoding *v-myc*. Scale bars = 200 μm.

**Figure 5 pharmaceuticals-16-00123-f005:**
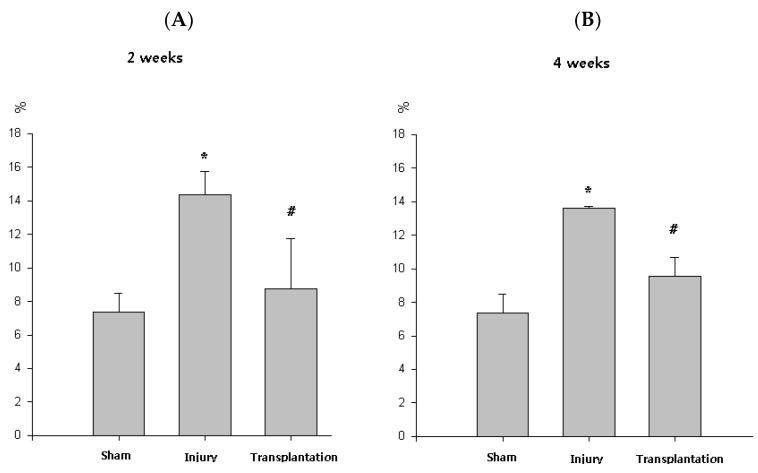
Percent collagen area of corpus cavernosa. (**A**,**B**) At 2, 4 weeks, the group with cavernous nerve injury showed significantly increased percent collagen area than the group without cavernous nerve injury (* *p* < 0.05). At 2, 4 weeks, the group transplanted with vREC showed significantly decreased percent collagen area than the group without vREC (# *p* < 0.05).

**Figure 6 pharmaceuticals-16-00123-f006:**
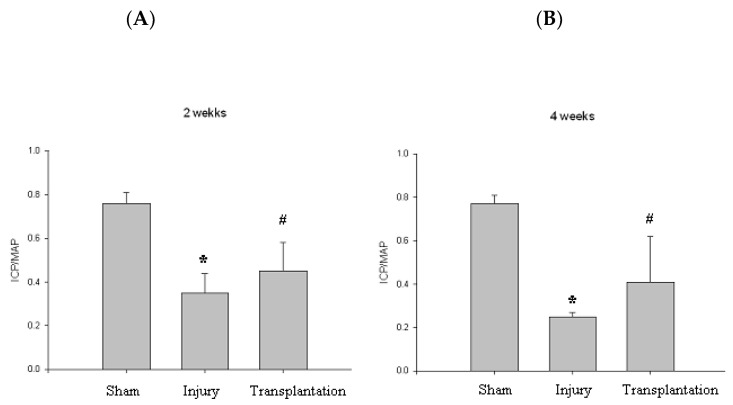
Erectile function. (**A**,**B**) At 2, 4 weeks, the group with cavernous nerve injury showed significantly lower ICP/MAP than the group without cavernous nerve injury (* *p* < 0.05). At 2, 4 weeks, the group transplanted with vREC showed higher ICP/MAP than the group without vREC (# *p* < 0.05). ICP/MAP = intracavernousal pressure/mean arterial pressure.

**Figure 7 pharmaceuticals-16-00123-f007:**
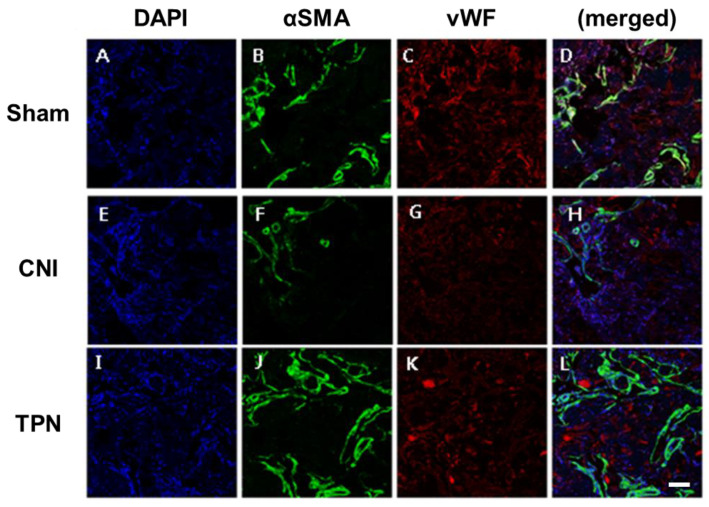
Expression of cavernous smooth muscle and endothelial makers after transplantation of vREC. (**A**–**D**) sham operation group, (**E**–**H**) cavernous nerve injury group (CNI), (**I**–**L**) vREC transplantation group (TPN). At 4 weeks, the group with cavernous nerve injury showed decreased density of expression of cavernous endothelial and smooth muscle makers than the sham operation group. Transplantation of vREC showed increased expression of cavernous endothelial and smooth muscle makers than cavernous nerve injury group. Sham = sham operation, CNI = cavernous nerve injury without vREC transplantation. TPN = cavernous nerve injury with vREC transplantation. DAPI = 4′,6-diamidino-2-phenylindole, SMA = smooth muscle actin, vWF = von Willebrand factor. Scale bars = 200 μm.

**Table 1 pharmaceuticals-16-00123-t001:** RT-PCR primers used.

Gene	Forward Primer	Reverse Primer	Tm (°C)
** *v-myc* **	GGTGTCACGTCAACATCCAC	GTTCGCCTCTTGTCGTTCTC	58
**CD31**	GGCCGAAGCTAGAACTCTCC	CACCTGGACGGGTACCAAAT	58
**CD105**	GTGTCTACATGGTGCCCACA	CCGATGCTGTGGTTGGTACT	58
**vWF**	TCAGTGTGTTGGGGACGATG	GCAAGTTGCAGTTGACCAGG	58
**Flt-1**	GAAAAGTCCGTGTCGTCCCT	GCTGAGTGATGCCCTCGATT	58
**KDR**	AAAGAGAGGGACTTTGGCCG	GTCGCCACTTGACAAAACCC	58
**GAPDH**	AGGTCGGTGTGAACGGATTTG	TGTAGACCATGTAGTTGAGGTCA	58

## Data Availability

Data is contained within the article.

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
