# Peer review of "Established Immortalized Cavernous Endothelial Cells Improve Erectile Dysfunction in Rats with Cavernous Nerve Injury"

_pharmaceuticals, 2023, doi:10.3390/ph16010123_

Round 1

Reviewer 1 Report

Dear authors, the topic of your work “Immortalized cavernous endothelial cells improve erectile dysfunction in rats with cavernous nerve injury ” it is very scientific importance in human clinical reproduction practice.

I would like the following considerations to the authors:

Simple summary and Abstract are corrected summarize. 

1. Introduction 

That is ok.

2. Materials and Methods 

In Measurement of erectile function: the sentences: “The penis was exposed, and the corpus spongiosum was mobilize to facilitate insertion of a 25-gauge needle into the corpus cavernosum=> Why is necessary this procedure?

3.- Results, discussion and Conclusions are very clear to read. 

In my opinion this is a very good work for publish in Pharmaceutics. 

I would like to congratulate the authors for their outstanding work on clinical application.

Author Response

Thank you for your careful comment. We established the cell line and transplanted it to different mice in this case, so this is an allogenic transplant for certain. However, various types of stem cell therapy for erectile dysfunction have been steadily proposed (Sex Med Rev, 2020, SD Lokeshwar, et al.; Sex Med Rev, 2016, TC Peak, et al.), and the results of using self-tissues in clinical trials of erectile dysfunction are quite encouraging. (Urology, 2018, MK Haahr, et al.; Urol Int, 2016, R Yiou, et al.) Using autologous cells has the advantage of not inducing the host's immune response and causing no inflammatory reaction. Therefore, what we wanted to express in the original context was to mention that our findings also have this possibility along with these main trends. Please note that we have tried to show general evidence that injecting v-myc immortalized cells results in improvement of impaired epithelial structure and function. But as you pointed out, the sentence can be misleading as if we were using autologous epithelial cells in this study, so we correct it as below.

: "Autologous endothelial cells can be transplanted into the corpus cavernosum." -> "We transplanted allogenic v-myc-transfected cells into the corpus cavernosum of rodents with CNI."

Reviewer 2 Report

In this study, the authors aimed to investigate immortalized rat penile cavernous ECs (rEC) could repair erectile dysfunction in rats with cavernous nerve injury (CNI). They reported that vREC showed the expression of CD31, vWF, cell type-specific markers for EC by RT-PCR and flowcytometry. At 2, 4 weeks after transplantation, rats with CNI had significantly lower erectile function than control group.

             This is an interesting article showing the efficacy of immortalized rat penile cavernous ECs for the treatment of CNI model, which mimic radical prostatectomy-induced erectile dysfunction. My comment on this article is somewhat minor.

1. In the first paragraph of discussion section, the authors mentioned that ‘autologous endothelial cells can be transplanted into the corpus cavernosum’. Have the authors used autologous rEC or allogeneic rEC in the present study? Please clarify.

2. It was also mentioned in discussion section that ‘The risk of myc is its tumorigenic potential, offsetting its great utility in promoting self-renewal. However, if myc’s tumorigenic potential could be reliably inactivated in vivo, it might be the only gene required to be invoked in vitro to enhance self-renewal of an endothelial progenitor, preserving its multipotence/ oligopotence without inducing pluripotence.’ Please explain in detail how myc’s tumorigenic potential could be reliably inactivated in vivo. And also, how the authors have confidence that immortalized EC do not exert tumorigenic potential? Please discuss more about this issues.

Author Response

#1. In the first paragraph of discussion section, the authors mentioned that ‘autologous endothelial cells can be transplanted into the corpus cavernosum’. Have the authors used autologous rEC or allogeneic rEC in the present study? Please clarify.

: Thank you for your careful comment. We established the cell line and transplanted it to different mice in this case, so this is an allogenic transplant for certain. However, various types of stem cell therapy for erectile dysfunction have been steadily proposed (Sex Med Rev, 2020, SD Lokeshwar, et al.; Sex Med Rev, 2016, TC Peak, et al.), and the results of using self-tissues in clinical trials of erectile dysfunction are quite encouraging. (Urology, 2018, MK Haahr, et al.; Urol Int, 2016, R Yiou, et al.) Using autologous cells has the advantage of not inducing the host's immune response and causing no inflammatory reaction. Therefore, what we wanted to express in the original context was to mention that our findings also have this possibility along with these main trends. Please note that we have tried to show general evidence that injecting v-myc immortalized cells results in improvement of impaired epithelial structure and function. But as you pointed out, the sentence can be misleading as if we were using autologous epithelial cells in this study, so we correct it as below.

: "Autologous endothelial cells can be transplanted into the corpus cavernosum." -> "We transplanted allogenic v-myc-transfected cells into the corpus cavernosum of rodents with CNI."

#2. It was also mentioned in discussion section that ‘The risk of myc is its tumorigenic potential, offsetting its great utility in promoting self-renewal. However, if myc’s tumorigenic potential could be reliably inactivated in vivo, it might be the only gene required to be invoked in vitro to enhance self-renewal of an endothelial progenitor, preserving its multipotence/ oligopotence without inducing pluripotence.’ Please explain in detail how myc’s tumorigenic potential could be reliably inactivated in vivo. And also, how the authors have confidence that immortalized EC do not exert tumorigenic potential? Please discuss more about this issues.

: In previous studies, we have tried to verify the different types of v-myc-transfected immortalized cells. (Mol Biol Rep., 2021, JH Kim, et al.; Cell Transplant., 2017, SS Choi, et al.) Some evidence has shown successful differentiation of v-myc-transfected cells in rodents (Exp Neurol., 2012, Park D, et al.; Proc Natl Acad Sci USA, 2011, Lee HJ, et al.; J Neurosci Res., 2010, Lee HJ, et al.), and v-myc has been a useful and safe oncogene for immortalized cells without tumorigenesis. (Cell Transplant., 2015, Lee SR, et al.) But as pointed out, it's not certain whether the tumorigenic potential of v-myc could be perfectly controlled in vivo. So we think of this as the term "could be", not "can be". The point we hope to focus on is the potential. Yet our findings do not try to verify the tumorigenic potential in vivo, but we have revealed that v-myc-transfected cells have the potential to form the proper structures and repair the functions both in vitro and in vivo. Surely further studies will be needed to ensure that controlled tumorigenicity and repairing capacity are available in vivo. We add some descriptions and references to our previous results for more information, as described above.